# Are There Heterogeneous Impacts of National Income on Mental Health?

**DOI:** 10.3390/ijerph17207530

**Published:** 2020-10-16

**Authors:** Zimei Huang, Tinghui Li, Mark Xu

**Affiliations:** 1School of Economics and Statistics, Guangzhou University, Guangzhou 510006, China; 1112064001@e.gzhu.edu.cn; 2Portsmouth Business School, University of Portsmouth, Portsmouth PO1 3DE, UK; Mark.Xu@port.ac.uk

**Keywords:** national income, mental health, heterogeneous impact, panel Tobit model

## Abstract

Understanding heterogeneous impact and mechanisms between national income and mental health are crucial to develop prevention and intervention strategies. Based on panel data from 2007 to 2017, this study explores the heterogeneous impact of national income on different types of mental health. Then, it analyzes the heterogeneous impact among countries with different income levels. Furthermore, the heterogeneous moderating effects of national income on mental health mechanisms are elaborated and the findings reveal several key conclusions: firstly, national income exerts a heterogeneous impact on different types of mental health. Rising national income is conducive to increase people’s happiness and reduce their prevalence of anxiety disorders, but it increases the prevalence of depression disorders. Secondly, national income has a heterogeneous impact on different types of mental health among countries with different income levels. Furthermore, the heterogeneous influence mechanism of national income on mental health is mainly reflected in different types of mental health. Unemployment, social support and freedom can moderate the relationship between national income and depression, while social support, positive affect and negative affect can moderate the relationship between national income and anxiety. Finally, based on the conclusions of quantitative analysis, some important policy recommendations are proposed for policy makers.

## 1. Introduction

In recent years, mental health conditions have been increasing worldwide. According to the World Health Organization (WHO), mental health is a state of well-being in which an individual realizes his or her own abilities, can cope with the normal stresses of life, work productively, and is able to make a contribution to his or her community. Mental health has a substantial effect on all areas of our daily life, such as school or work performance, relationships with family and friends, ability to participate in the community and physical health. Mental health diseases represent a growing segment of the global burden of disease [1,2]. Two of the most common mental health conditions, depression and anxiety [3,4], which are highly prevalent and disabling disorders, result not only in an enormous amount of human misery and loss of health, but also in a reduction in economic output [5]. For example, they cost the global economy USD 1 trillion each year. Therefore, there is a growing recognition that mental health plays an important role in achieving global development goals, as well as illustrating by the inclusion of mental health in the Sustainable Development Goals. In order to reduce mental health diseases, it is important to understand the underlying factors and mechanisms that may influence mental health. This may help to provide tools for prevention and intervention strategies.

Extant studies on mental health have primarily focused on its influencing factors. Various types of mental health include happiness (subjective wellbeing), schizophrenia, depression and anxiety disorders and so on. There are specific psychological and personality factors that make people vulnerable to mental health problems. On the one hand, various studies have analyzed the individual, familial and societal determinants of mental health, with the literature identifying the following factors: gender [6,7], genetic factors, financial difficulties [8], social policy, social support [9], environment [10], and education [11,12]. For example, Richardson, et al. [13] found that greater financial difficulties predicted greater depression and stress cross-sectionally, and also predicted poorer anxiety, global mental health and alcohol dependence over time. Butterworth, et al. [8] investigated the association between depression and financial hardship over time and discovered that current financial hardship was strongly and independently associated with depression.

On the other hand, socio-economic risk factors for mental health problems are economic crisis [14,15,16,17,18], credit cycle [19], poverty [20], income inequality [21,22,23,24], unemployment [25], socioeconomic inequalities [26,27], socio-economic status [28,29,30] and so on [31,32,33,34]. For instance, Bartoll, et al. [16] showed the deterioration in mental health among men during the economic crisis in Spain, especially among those from a low socioeconomic position. Gili, et al. [35] indicated a substantial increase in the prevalence of most types of mental health disorders, especially for depression, among primary care attendees between 2006 and 2010 during the period of economic crisis, with unemployment and mortgage payment difficulties as major risk factors. Katikireddi, et al. [36] assessed short-term differences in population mental health before and after the 2008 recession and explored how and why these changes differ by gender, age and socio-economic position. The results implied that no clear evidence for an increase in inequalities is associated with the recession. Jenkins, et al. [20] found that both low income and debt are associated with mental illness, but the effect of income appears to be mediated largely by debt. Similarly, no evidence was found for a differential effect between age groups. Kahn, et al. [37] examined the association of state income inequality and individual household income with the mental and physical health of women with young children; the results indicated that high income inequality confers an increased risk of poor mental and physical health. Both income inequality and household income are important for health.

Existing literature concerning the impact of national income on mental health mainly focuses on specific types of mental health. For example, Sarracino [38] investigated the determinants of subjective well-being in high- and low-income countries and the results indicated that happiness equations are the same across countries. De Neve and Oswald [39] used sibling fixed effects to estimate the influence of life satisfaction and positive affect on later income. The results implied that happiness and income are connected by a two-way relationship, and that human well-being can itself be a source of economic dynamism. Yatham, et al. [1] reviewed the published data on the prevalence and randomized trials of interventions for depression, anxiety and post-traumatic stress disorder in youth in low- and middle-income countries. They found that the mental health burden due to depression and anxiety disorders in youth is substantial in low- and middle-income countries, with high needs but inadequate services.

However, previous studies evaluating the impact of national income on mental health have not paid much attention to its potential heterogeneous effect on mental health. We are trying to fill the gap by investigating the heterogeneous impact of national income on mental health, including various countries, different types of mental health and their various influencing mechanisms.

From above, this study makes three contributions to the existing literature on national income and mental health. Firstly, employing the panel Tobit model, this paper expands on the current literature on exploring the heterogeneous impact of national income on different types of mental health. These types of mental health are divided into happiness, depression and anxiety disorder. The results indicate that national income exerts a heterogeneous impact on mental health, which mainly reflects on different types of mental health. Specifically, raising the national income is beneficial to improve our happiness and relieve our anxiety, but it also will increase the prevalence of depression.

Secondly, this paper attempts to investigate the heterogeneous impact of national income on mental health under various countries. According to the country classifications of World Bank, this paper divides countries into three types: high-, middle- and low- income countries. The results imply that heterogeneous impact of national income on different types of mental health reflected in countries with different income levels. Concretely, for high-income countries, national income has an insignificant impact on happiness, while it has an inverted “U” shaped relationship with depression and anxiety. In middle-income countries, national income has an inverted “U” shaped relationship with three types of mental health (happiness, depression and anxiety). In low-income countries, national income exerts a significantly negative impact on happiness, while it exerts a significantly positive effect on depression. At the same time, national income has an inverted “U” shaped relationship with anxiety. These suggest that different income level countries should pay attention to their own national mental health and formulate corresponding policies.

Finally, this paper tries to further analyze the heterogeneous influencing mechanism between national income and mental health. The results show that the heterogeneous influence mechanism of national income on mental health is mainly reflected in different types of mental health. For example, unemployment, social support and freedom play moderating effects on national income depression mechanisms, while social support, positive and negative effect play moderating effects on national income anxiety mechanisms.

The remainder of this paper is organized as follows (see Figure 1 for the logical framework). Section 2 develops the hypotheses and introduces method and data. Section 3 empirically examines the heterogeneous impact of national income on different types of mental health. Section 4 explores the heterogeneous impact of national income on mental health among countries with different income levels. Section 5 further studies the moderating effect on national income–mental health mechanisms. The conclusions and policy implications are shown in Section 6.

## 2. Hypotheses, Method and Data

### 2.1. Hypotheses

As the WHO has stressed, mental health is about not only the absence of mental disorders or disabilities, but also looking after ongoing wellness and happiness. Therefore, the most common types of mental health include happiness, depression [8,24] and anxiety disorder [1,20,27]. Firstly, national income has a positive impact on happiness. Income matters in terms of happiness by helping to meet basic needs as well as sustaining well-being in times of economic shocks and crises [7,40]. High income offers a much better chance of getting a better education and accessing better nutrition and medical care, which minimize the impact on their enjoyment of life. At the same time, higher incomes lead to a greater sense of accomplishment and security, which makes people happy. Secondly, national income exerts a positive impact on depression. People with higher incomes tend to spend more time at work, they are often stressed for longer periods of time, and do not spend much time on particularly enjoyable activities to relieve stress, thus leading to higher prevalence of depression disorder. Thirdly, national income has a negative effect on anxiety. People with higher incomes are more likely to meet their own needs and be relatively satisfied with their quality of life, thus helping them to reduce anxiety. To sum up, national income has a heterogeneous effect on various types of mental health. Therefore, it is reasonable to hypothesize that:

**Hypotheses** **1.**
*The heterogeneous impact of national income on mental health is mainly reflected in different types of mental health.*


National income is closely related to people’s own survival and needs development. Income provides the basic conditions of existence necessities such as food, clothing and shelter to meet basic human needs, increases happiness, and reduces the prevalence of depression and anxiety disorders. At the same time, income can also help people achieve security, status and the development of their abilities to meet self-actualization. In high-income countries, higher incomes can provide people with more material wealth and more favorable opportunities and choices, and satisfy more of their needs, thus, it is beneficial to their mental health. In middle-income countries, national income has an inverted “U” shaped relationship with mental health. People with very low incomes and very high incomes have relatively low levels of happiness, while those with middle incomes have the highest levels of happiness. In low-income countries, inadequate survival necessities, poor nutrition and medical conditions increase the likelihood of illness and disability, and thus can be bad for people’s mental health. In this vein, it could stand to reason that there are heterogeneous effects of national income on different types of mental health in countries with different income levels. Therefore, we formulate the following hypothesis:

**Hypotheses** **2.**
*The heterogeneous effect of national income on different types of mental health is mainly reflected in countries with different income levels.*


Mental health is affected by many factors other than national income, such as unemployment, social support, freedom, generosity, positive affect, negative affect and so on. The increase in national income can reduce the unemployment and negative affect, as well as increase social support, freedom, generosity and positive affect, thus, it is further conducive to improve people’s mental health. Specially, unemployment, social support and freedom have moderating effects on national income–depression mechanisms. The rise of unemployment will lead to a decline of social support and freedom and the social support and freedom are beneficial to increasing the relationship between social support and freedom. Social support, positive and negative effects have moderating effects on national income–anxiety mechanisms. As mentioned above, because of the various types of mental health, the influence mechanisms of national income on mental health may be heterogeneous. Therefore, we hypothesize that:

**Hypotheses** **3.**
*The heterogeneity of the influence mechanism of national income on mental health is mainly reflected in different types of mental health.*


### 2.2. Panel Tobit Model

Based on the above theoretical analysis, the panel Tobit model could be regarded as an effective tool to investigate the impact of national income on mental health. Happiness, the prevalence of depression and anxiety disorder are characterized by non-negative truncation. For the estimation of such a constrained dependent variable model, the ordinary least square (OLS) method obtains biased results, so it is not suitable to use the OLS for coefficient estimation. However, the panel Tobit model estimated by the maximum likelihood (ML) method is an alternative to the OLS method [41]. The Tobit model is also called the sample selection model or limited dependent variable model [42]. In panel Tobit model, the change of dependent variable is limited to some extent [43], and variables with limited values are defined as “deletion” or “truncation”. In this paper, happiness, the prevalence of depression and anxiety disorder are always limited to no less than 0. So, the panel Tobit model is more appropriate. At the same time, the fixed effect Tobit model is usually unable to obtain consistent and unbiased estimators, while the random effect model is better. Therefore, we specify the random effect panel Tobit models as follows:(1)$$\begin{array}{cc} MH_{it}= & \alpha +\beta_{1}*NI_{it}+\beta_{2}*Unemp_{it}+\beta_{3}*Soci_{it}+\beta_{4}*Free_{it}\\ & +\beta_{5}*Gene_{it}+\beta_{6}*Posi_{it}+\beta_{7}*Nega_{it}+\varepsilon_{it} \end{array},$$
(2)$$\begin{array}{cc} Happ_{it}= & \alpha +\beta_{1}*NI_{it}+\beta_{2}*Unemp_{it}+\beta_{3}*Soci_{it}+\beta_{4}*Free_{it}\\ & +\beta_{5}*Gene_{it}+\beta_{6}*Posi_{it}+\beta_{7}*Nega_{it}+\varepsilon_{it} \end{array},$$
(3)$$\begin{array}{cc} Dep_{it}= & \alpha +\beta_{1}*NI_{it}+\beta_{2}*Unemp_{it}+\beta_{3}*Soci_{it}+\beta_{4}*Free_{it}\\ & +\beta_{5}*Gene_{it}+\beta_{6}*Posi_{it}+\beta_{7}*Nega_{it}+\varepsilon_{it} \end{array},$$
(4)$$\begin{array}{cc} Anx_{it}= & \alpha +\beta_{1}*NI_{it}+\beta_{2}*Unemp_{it}+\beta_{3}*Soci_{it}+\beta_{4}*Free_{it}\\ & +\beta_{5}*Gene_{it}+\beta_{6}*Posi_{it}+\beta_{7}*Nega_{it}+\varepsilon_{it} \end{array},$$
where MH represents the mental health; Happ,  Dep and Anx stands for happiness, prevalence of depression and anxiety disorder, respectively; NI refers to national income; Unemp is unemployment; Soci represents the social support; Free is freedom; Gene stands for generosity; Posi refers to positive affect; Nega represents negative affect; α is the constant term; β is the regression coefficient; ε is the random disturbance term.

### 2.3. Variables Selection and Data Source

This paper mainly focuses on the heterogeneous effect of national income on mental health in the period 2007–2017. The main reasons for selecting the sample period 2007–2017 are that the data of prevalence of depression and anxiety disorder after 2017, which were obtained from the Global Burden of Disease Study (GBD), have not been published. The GBD provides a comprehensive assessment of all-cause and cause-specific mortality for 249 causes in 195 countries and territories from 1980 to 2017. These results informed an in-depth investigation of observed and expected mortality patterns based on socio-demographic measures [44]. Considering the availability and completeness of data, this paper selects 55 countries as samples. According to the classification of World Bank, we divide the sample into three sub-samples: high-, middle- and low- income countries. The high-income countries include 15 countries (Canada, Chile, Germany, Denmark, Spain, United Kingdom, Israel, Italy, Japan, Korean Rep., Lithuania, Panama, Sweden, Uruguay, United States). The middle-income countries sample includes 36 countries (Argentina, Armenia, Belarus, Brazil, China, Colombia, Costa Rica, Dominican Republic, Ecuador, Guatemala, Kazakhstan, Mexico, Peru, Paraguay, Turkey, Russian Federation, Thailand, South Africa, Bangladesh, Bolivia, Cameroon, Egypt, Ghana, Honduras, India, Kenya, Kyrgyzstan, Cambodia, Moldova, Mauritania, Nicaragua, Pakistan, Philippines, Senegal, El Salvador, Ukraine), while the low-income countries sample includes 4 countries (Niger, Nepal, Tanzania, Uganda).

All data are obtained from the Gallup World Poll (GWP), Global Burden of Disease (GBD), World Bank (WB) and World Development Indicators (WDI) database. The explained variable of the panel Tobit model is mental health (including happiness, prevalence of depression and anxiety disorders), the prevalence of depression and anxiety disorders is defined by the number of people in the sample with the characteristic of depression and anxiety disorders, divided by the total number of people in the sample, respectively [45]. National income is the explanatory variable. Moreover, to avoid an omitted variable bias, certain related control variables are included in our model. The control variables added to the model include six variables: unemployment, social support, freedom to make life choices, generosity, positive and negative affect. The measurements and sources of all variables are shown in Table 1.

### 2.4. Descriptive Statistics

The descriptive statistics of variables are presented in Table 2. Descriptive statistics are presented to describe the basic characteristics of data in this study concerning 55 countries in the period 2007–2017. For each variable, we present the mean, standard deviation (Std. Dev.), minimum (Min) and maximum (Max). As shown in Table 2, there are significant divergences on the range of mental health among different income level countries. Firstly, in the high-income countries, happiness ranges from 5.066 to 7.971, depression ranges from 2.586 to 7.971, while anxiety ranges from 2.882 to 7.209. Secondly, in the middle-income countries, happiness ranges from 3.559 to 7.615, depression ranges from 2.159 to 5.434, while anxiety ranges from 2.472 to 6.682. Thirdly, in the low-income countries, happiness ranges from 2.903 to 5.099, depression ranges from 2.142 to 3.511, while anxiety ranges from 2.401 to 3.786. Besides that, we focus on the average national income in different countries. Specifically, the average national income in high-income countries is 3.434. For the middle-income countries, the average national income is 0.441. The average national income in low-income countries is 0.061.

Figure 2 shows the relationship between national income and different types of mental health. Figure 2a implied that people from high-income countries are happier than people from low-income countries. This is in line with the study of CUMMINS [46], which suggests that people who are poor will experience lower happiness than people who are rich. There is an obvious positive relationship between national income and happiness. In other words, the higher the national income, the happier people are. To some extent, increasing national income is crucial to improving people’s happiness. Figure 2b showed that there exists a positive relationship between national income and depression. Figure 2c preliminarily demonstrated that there is no significant correlation between national income and anxiety. However, Figure 2 is only a preliminary depiction of typical facts. It is necessary to take other factors affecting mental health into comprehensive consideration to obtain a more reliable conclusion and further analyze the heterogeneous impact of national income on mental health.

## 3. The Heterogeneous Impact of National Income on Different Types of Mental Health

This section discusses regression results on the relationship between national income and mental health. First, we analyze the stationarity of variables by results of panel unit root test in Section 3.1. Then we employ the panel Tobit model to explore the heterogeneous effects of national income on different types of mental health in Section 3.2.

### 3.1. Stationarity Test

We perform a panel unit root test to check whether all variables are stationary before we test the impact of national income on mental health. Table 3 reports the results for a battery of panel unit root tests for happiness, depression, anxiety, national income, unemployment, social support, freedom, generosity, positive and negative affect. In particular, we report results from all tests of the null hypothesis that each series contains a unit root. The first is the Levin–Lin–Chu unit-root test [47]. The second is the Fisher type augmented Dickey–Fuller test [48,49]. The third one is the Im–Pesaran–Shin test [50], and the forth is the Harris Tzavalis test [51]. The decision criterion is that the variable is stationary if the unit root tests confirm non-rejection of the null at a 5% level of significance [52]. In the first three tests cases, the tests reject the null hypothesis at the 1% level of significance. Consequently, we can conclude that all variables are stationary.

### 3.2. The Heterogeneous Impact on Different Types of Mental Health

In this part, our study explores the heterogeneous impact of national income on different types of mental health. Thus, we employ the panel Tobit model to estimate the regression coefficients. Happiness, depression, and anxiety are the explained variables, respectively, while national income is the explanatory variable, and unemployment, social support, freedom, generosity, positive and negative affect are used as control variables. Columns 2, 3 and 4 in Table 4 present the results of the panel Tobit model. The rest of the columns in Table 4 present the results of the panel regression model for robustness check. As seen in Table 4, the results are conclusive among different methods used.

National income exerts a heterogeneous impact on mental health, and the heterogeneous impact mainly focuses on different types of mental health. It is apparent from the results of panel Tobit model and panel regression model given in Table 4 that national income has a heterogeneous impact on mental health, which confirms the Hypothesis 1 of our study. National income is closely related to people’s own survival and development needs. Income provides the basic conditions of existence necessary such as food, clothing and shelter necessary to meet basic human needs, increases happiness, and reduces prevalence of depression and anxiety disorders. At the same time, income can also help people achieve security, status and the development of their abilities to meet self-actualization.

Firstly, national income has a significantly positive impact on happiness. As shown in the second column of Table 4, the coefficient of the impact of the national income on happiness is 0.352, which is significant at the 1% level; the results of panel regression model also show the same result. This result suggests that increasing national income is beneficial to improve our happiness. That is because people with higher incomes tend to have better health and mental health, have greater longevity, are less frequently the victims of violent crime, and experience fewer stressful life events. In addition, richer people score higher in characteristics such as interpersonal trust. Therefore, national income exerts a significantly positive effect on happiness.

Secondly, national income exerts a significantly positive impact on depression. The third column of Table 4 showed that the coefficient of the impact of the national income on depression is 0.233, which is significant at the 1% level. That is to say, increasing national income will increase the prevalence of depression disorders. The result is somewhat consistent with the results of CUMMINS [46], who discovered the dominance of financial concerns among low-income people and its replacement by higher-order concerns as income rises to meet basic needs. For example, higher incomes over time may also be related to increased pollution [10], congestion, stress, or other negative influences that may cause the prevalence of depression disorders to rise with income. This result provides a shred of evidence that with the increase in national income, people’s life pressure is increasing day by day. People tend to spend more time to work, and less time to communicate with family and friends and relax, which results in stress and negative emotions can not be dealt with in time, leading to depression [53]. Besides that, with the increase in national income, air pollution increases, and this will reduce hedonic happiness and increase the prevalence of depression disorders. Therefore, the increase in national income will lead to a larger prevalence of depression disorders.

Thirdly, national income has a significantly negative impact on anxiety. As shown in the fourth column of Table 4, the coefficient of the impact of the national income on anxiety is −0.051, which is significant at the 1% level. In other words, increasing national income is conducive to reduce the prevalence of anxiety disorders. One of the explanations of this phenomenon is that higher national income can bring about more material satisfaction, and people do not need to worry more about their basic needs in life, thus easing their anxiety. That is to say, as national income rises, people’s lives are guaranteed, thus reducing their anxiety.

To sum up, national income exerts a heterogeneous impact on mental health, and the heterogeneous impact mainly reflects different types of mental health.

## 4. The Heterogeneous Impact among Countries with Different Income Levels

In this section, we further explore the heterogeneous impacts of national income on different types of mental health among countries with different income levels. We also employ the panel Tobit model to estimate the coefficients, and the results are presented in Table 5, Table 6 and Table 7.

National income has a heterogeneous impact on different types of mental health, and the heterogeneous impact is mainly reflected in countries with different income levels. From the results of the panel Tobit model given in Table 5, Table 6 and Table 7, we can conclude that national income has a heterogeneous impact on different types of mental health among countries with different income levels, which confirms the Hypothesis 2 of our study.

Firstly, in high-income countries, national income has an insignificant impact on happiness. The result is in line with the results of Kahneman and Deaton [54], who argued that high income buys life satisfaction but not happiness. This result is also consistent with the results of Diener [55], which indicate that as real income increases within a country, people do not necessarily report more happiness. The reasons for the insignificant impact may be that people who are rich, tend to pursue higher material goals and other values. They must earn their money, and might be required to spend more of their time in work, and have less time available for leisure (such as exercise, shopping and childcare) and social relationships [53,56]. Thus, they tend to be less happy and experience more tension and stress. At the same time, rich people might adapt to their conditions, and have rising expectations and desires that counteract the effects of the desirable circumstances of their lives. Income appears to increase happiness little over the long-term when material desires of rich people rise with their incomes. Besides that, higher incomes over time may also be related to increased environmental pollution, stress, or other negative influences that may prevent happiness from rising with income. As a result, higher incomes might create lower feelings of well-being.

As for depression, when we only consider the linear relation between national income and depression, national income has a positive effect on depression. However, when we take nonlinear relationships into consideration, national income has an inverted “U” shaped relationship with depression. In other words, when national income reached the threshold value (0.0394 = −(−0.042/2 * 0.533)), the prevalence of depression no longer rose with national income, but fell. This inverted “U” shaped relationship may be closely related to the phenomenon: the rich need to spend more time on work to pursue higher social status, wealth and power, and it may result in greater pressure. The pressure will increase the prevalence of depression disorder. However, when the national income increased and reached a threshold, the prevalence of depression decreased.

As for anxiety, when we only consider the linear relation between national income and anxiety, national income has a negative effect on anxiety. However, when we take the nonlinear relationship into consideration, national income has an inverted “U” shaped relationship with anxiety. In other words, when national income reaches the threshold value (2.357 = −(0.099/2 * (−0.021)), the prevalence of anxiety no longer rises with national income, but falls. This may because when the rich have not met their expectations, they will try their best to pursue them, which will bring more anxiety. However, when they reach their goals (for example, earn enough money), their anxiety will reduce.

According to the above analysis, when the national income reaches 2.357, either the prevalence of depression or anxiety increased with the rising national income.

Secondly, in middle-income countries, national income has an inverted “U” shaped relationship with three types of mental health. In other words, there is a national income threshold, above which happiness and prevalence of depression and anxiety disorder decrease with the increase in national income, and below which happiness and prevalence of depression and anxiety disorder increase with the increase in national income. The happiness will increase when national income increases. However, when national income reaches the threshold (1.254 = −2.723/(2 × −1.086)), the happiness is no longer rising with national income, but is falling. That is to say, people with very low incomes and very high incomes have relatively low levels of happiness, while those with middle incomes have the highest levels of happiness. As for depression and anxiety, the prevalence increases with the rise of national income; when national income reaches the threshold (1.078 = −1.609/(2 × −0.746)) or (0.893 = −0.475/(2 × −0.266)), the prevalence of depression and anxiety disorder decreases with the increase in national income, respectively.

Based on the above analysis, when national income is between 1.078 and 1.254, happiness rises with increasing national income. At the same time, the prevelence of depression and anxiety decreases with the rising national income. In other words, when national income ranges from 1.078 to 1.254, people have the healthiest mental health in middle-income countries.

Thirdly, in low-income countries, national income has a significantly negative impact on happiness, while national income exerts a significantly positive effect on depression. The reasons for these two phenomena may be that low national income may not provide people with enough material wealth and adequate survival necessities, and satisfy their needs. Their lack of income reflects a general lack of resources and the experience of an environment. At the same time, poor nutrition and medical conditions can increase the likelihood of illness and disability, and thus can be detrimental to their mental health, reduce their happiness and increase the prevalence of depression disorders. Besides that, national income has an inverted “U” shaped relationship with anxiety. When national income reaches the threshold value (0.111 = −17.56/(2 × −79.39)), the prevalence of anxiety disorders decreases with the increase in national income.

In conclusion, national income has heterogeneous impact on different types of mental health, and the heterogeneous impact is mainly reflected in countries with various income levels.

## 5. The Heterogeneous Moderating Effects on National Income–Mental Health Mechanisms

In this section, we further investigate the heterogeneous moderating effects on national income–mental health mechanisms. As pointed out in the moderating hypothesis of this study, moderating variables will moderate the relationship between the national income and different types of mental health. Thus, we would expect to test a moderating effect between the national income and different types of mental health considering different moderating variables. In further research, we therefore investigate the relationship between national income and different types of mental health because of the moderating hypothesis of our study. For this purpose, we estimate the following moderating effect model:(5)$$MH_{it}=\alpha +\beta *NI_{it}+\gamma_{1}*Moderate_{it}+\gamma_{2}*NI_{it}*Moderate_{it}+\lambda *Control_{it}+\varepsilon_{it},$$
(6)$$Happ_{it}=\alpha +\beta *NI_{it}+\gamma_{1}*Moderate_{it}+\gamma_{2}*NI_{it}*Moderate_{it}+\lambda *Control_{it}+\varepsilon_{it},$$
(7)$$Dep_{it}=\alpha +\beta *NI_{it}+\gamma_{1}*Moderate_{it}+\gamma_{2}*NI_{it}*Moderate_{it}+\lambda *Control_{it}+\varepsilon_{it},$$
(8)$$Anx_{it}=\alpha +\beta *NI_{it}+\gamma_{1}*Moderate_{it}+\gamma_{2}*NI_{it}*Moderate_{it}+\lambda *Control_{it}+\varepsilon_{it},$$
where MH represents the mental health; Happ,  Dep and Anx stands for happiness, prevalence of depression and anxiety disorder, respectively; NI refers to national income; Moderate is the moderating variable; $NI_{it}*Moderate_{it}$ represents the interaction term; Control is a vector of control variables, including unemployment, social support, freedom to make life choices, generosity, positive and negative affect; α is the constant term; $\beta ,\gamma_{1},\gamma_{2},\lambda$ stand for the regression coefficient of national income, moderating variable, interaction term and control variables, respectively; ε is the random disturbance term.

Before the moderating effect model is estimated, the data need to be processed. The observed values of the explanatory variable (national income) and the moderating variables minus their arithmetic averages respectively. Table 8, Table 9 and Table 10 present the results of the moderating effect between national income and mental health. The heterogeneity of the influence mechanism of national income on mental health is mainly reflected in different types of mental health. Firstly, from the Table 8, we find that all six moderating variables have an insignificant moderating effect on the relationship between national income and happiness.

Secondly, unemployment has a positive moderating effect on the relationship of national income and depression, while social support and freedom have a negative moderating effect on the relationship between national income and depression. From the Table 9, the interaction terms of the national income and unemployment, social support and freedom are 0.294, −0.242 and −0.149, respectively. They are statistically significant at the 1% level. The reason may be that the increase in unemployment will increases the life stress of people and fails to meet the necessary conditions of life, increasing the prevalence of depression; thus, unemployment has a positive moderating effect on the relationship between national income and depression. In addition, the social support and freedom are conducive to reduce the negative effect of national income on the prevalence of depression. In other words, the increase in social support and freedom reduce the prevalence of depression. People with satisfactory social support are more likely to be happy, less sad, improve quality of life, and reduce the prevalence of depression. As for freedom, economic freedom can meet the needs of people’s lives and reduces pressure and worries from the debt, which can also be beneficial to reduce the prevalence of depression.

Thirdly, social support and positive affect have a positive moderating effect on the relationship between national income and anxiety, while negative affect has a negative moderating effect on the relationship between national income and anxiety. From the Table 10, the interaction terms of the national income and social support, positive affect and negative affect are 0.083, 0.099 and −0.164, respectively. They are statistically significant at the 10%, 1% and 1% level, respectively. The results indicated that positive affect is conducive to increase the positive effect of national income on the prevalence of anxiety, and the negative affect reduces the negative effect of national income on the prevalence of anxiety.

All in all, the influence mechanisms of national income on mental health are heterogeneous, which is mainly reflected in different types of mental health.

## 6. Conclusions

The main objective of this study is to investigate the heterogeneous impact of national income on mental health. Employing a panel Tobit model, firstly, we explore the heterogeneous impact of national income on different types of mental health in the period 2007–2017. Secondly, we explore the heterogeneous impact of national income on mental health among countries with different income levels by dividing 55 countries into the high-, middle- and low- income countries sub-samples. Furthermore, we further investigate the heterogeneous moderating effects on national income–mental health mechanisms.

Based on the empirical results, several important conclusions are drawn as follows. Firstly, national income exerts a heterogeneous impact on different types of mental health. Specifically, national income has a significantly positive impact on happiness and depression, while it exerts a significantly negative impact on anxiety. In other words, increasing national income is conducive to increase people’s happiness and reduce the prevalence of anxiety disorders, but increase the prevalence of depression disorders.

Secondly, national income has heterogeneous impact on different types of mental health among countries with different income levels. To be specific, in high-income countries, national income has an insignificant impact on happiness, while national income has an inverted “U” shaped relationship with depression and anxiety. In middle-income countries, national income has an inverted “U” shaped relationship with three types of mental health (happiness, depression and anxiety). In low-income countries, national income has a significantly negative impact on happiness, while it exerts a significantly positive effect on depression. At the same time, national income has an inverted “U” shaped relationship with anxiety.

Finally, the heterogeneous influence mechanism of national income on mental health is mainly reflected in different types of mental health. Concretely, all six moderating variables have an insignificant moderating effect on the relationship between national income and happiness. However, unemployment has a positive moderating effect on the relationship between national income and depression, while social support and freedom have a negative moderating effect on the relationship between national income and depression. Moreover, social support and positive affect show a positive moderating effect on the relationship between national income and anxiety, while negative affect has a negative moderating effect on the relationship between national income and anxiety.

Accordingly, the following policy implications can be pursued to improve mental health. Firstly, national mental health policies should be concerned about different types of mental disorders; for example, improving the social support to prevent people suffering from anxiety. Besides that, as countries with different income levels have different social systems and social structures, appropriate policies should be formulated in accordance with the mental health conditions of their people. For example, treatment for mental disorders is limited in low- and middle-income countries. Therefore, they should strengthen public mental health facilities and corresponding treatment capacity. A series of interventions for the prevention, treatment and care of mental health problems, most of which can be delivered through community and routine health care platforms, use task-sharing by non-specialist providers [22]. Furthermore, mental health promotion should be mainstreamed into governmental and nongovernmental policies and programs. In addition to the health sector, it is essential to involve the education, labor, justice, transport, environment, housing, and welfare sectors. People’s mental health can be effectively improved only when all departments are organized to form a unified whole.

Our work is not without limitations, some of which open new avenues for future research. First, this paper has not taken age, gender, cultural differences and other factors into consideration. Second, this article does not distinguish between long-term or short-term effects on the relationship between the national income and mental health. Third, happiness in this paper is measured on a cardinal scale. The happiness scores are the national-level average scores in a country/region.

Thus, more future studies are needed. As a first extension, further research could take age, gender, cultural difference and other factors into account and analyze the heterogeneous effects of national income on mental health between these factors. As a second extension, future research could examine the short- and long-term effects of the national income on mental health. Third, despite the use of comparable methods across the countries, it is possible that people in different countries may conceptualize and answer survey questions differently. Therefore, measurement biases, both in terms of the mental health and its determinants, cannot be ruled out. (See Table A1, Table A2, Table A3 and Table A4 about the correlation matrices among variables in Appendix A).

## Figures and Tables

**Figure 1 ijerph-17-07530-f001:**
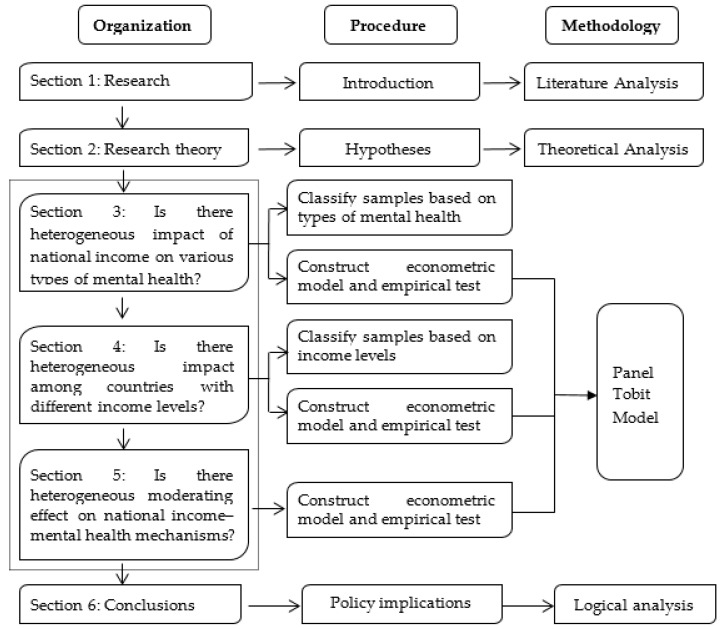
The logical organization of this paper.

**Figure 2 ijerph-17-07530-f002:**
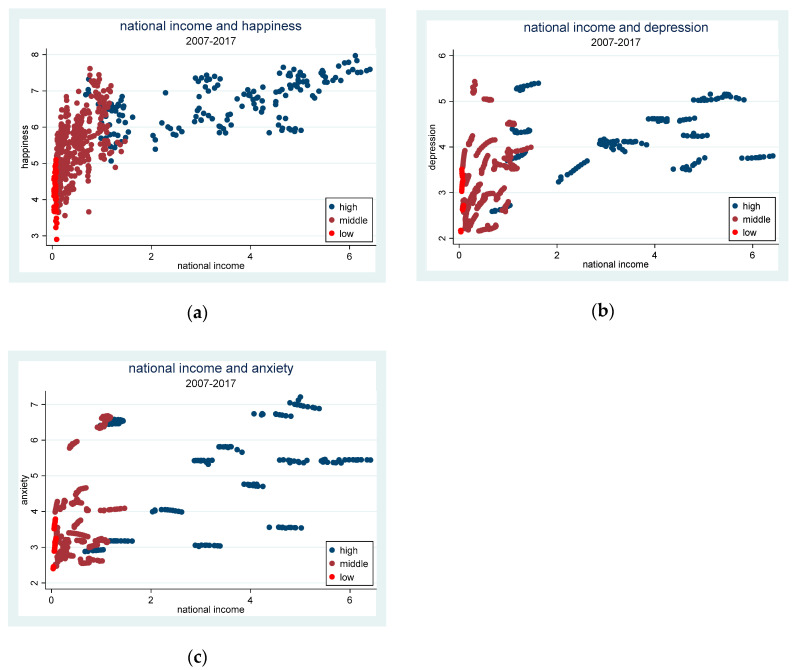
The relationship between national income and different types of mental health. (**a**) The relationship between national income and happiness. (**b**) The relationship between national income and depression. (**c**) The relationship between national income and anxiety. The sample period is from 2007 to 2017.

**Table 1 ijerph-17-07530-t001:** Variables and Data Source.

Nature of Variables	Variables	Measurement	Source
Dependent variable	Happiness	National average response to the question of life evaluations	GWP
	Depression disorders	Prevalence of depression disorders	GBD
	Anxiety disorders	Prevalence of anxiety disorders	GBD
Independent variable	National income	per capita values for gross national income (constant 2010 USD)	WB
Control variable	Unemployment	Unemployment, total (% of total labor force)	WDI
	Social support	National average of the binary responses (either 0 or 1) to the GWP question	GWP
	Freedom to make life choices	National average of responses to the GWP question	GWP
	Generosity	Residual of regressing national average of response to the GWP question on GDP per capita	GWP
	Positive affect	Average of three positive affect measures in GWP: happiness, laughter and enjoyment in the Gallup World Poll waves 3–7	GWP
	Negative affect	Average of three negative affect measures in GWP. They are worry, sadness and anger	GWP

Notes: the measurements shown in this table all refer to the World Happiness Report 2020. GWP denotes Gallup World Poll; GBD denotes Global Burden of Disease (University of Washington); WB denotes World Bank; WDI denotes World Development Indicators database.

**Table 2 ijerph-17-07530-t002:** Descriptive statistics.

Variables	Sample	Obs.	Mean	Std. Dev.	Min	Max
Happiness	Full	605	5.646	1.042	2.903	7.971
	1	165	6.685	0.640	5.066	7.971
	2	396	5.376	0.837	3.559	7.615
	3	44	4.181	0.495	2.903	5.099
Depression	Full	605	3.410	0.868	2.142	5.434
	1	165	4.186	0.681	2.586	5.397
	2	396	3.151	0.762	2.159	5.434
	3	44	2.844	0.489	2.142	3.511
Anxiety	Full	605	3.912	1.295	2.401	7.209
	1	165	5.042	1.356	2.882	7.209
	2	396	3.536	1.012	2.472	6.682
	3	44	3.056	0.453	2.401	3.786
National income	Full	605	1.229	1.632	0.033	6.413
	1	165	3.434	1.668	0.675	6.413
	2	396	0.441	0.336	0.0701	1.469
	3	44	0.061	0.017	0.033	0.092
Unemployment	Full	605	0.063	0.045	0.003	0.271
	1	165	0.075	0.043	0.023	0.261
	2	396	0.064	0.046	0.004	0.271
	3	44	0.018	0.009	0.003	0.036
Social support	Full	605	0.828	0.100	0.373	0.975
	1	165	0.906	0.045	0.738	0.975
	2	396	0.802	0.102	0.373	0.959
	3	44	0.771	0.072	0.582	0.885
Freedom	Full	605	0.744	0.131	0.335	0.970
	1	165	0.786	0.124	0.496	0.970
	2	396	0.731	0.132	0.335	0.964
	3	44	0.700	0.101	0.413	0.880
Generosity	Full	605	−0.002	0.156	−0.311	0.549
	1	165	0.064	0.167	−0.302	0.404
	2	396	−0.037	0.144	−0.311	0.549
	3	44	0.071	0.106	−0.065	0.323
Positive	Full	605	0.741	0.100	0.450	0.944
	1	165	0.765	0.087	0.473	0.893
	2	396	0.738	0.106	0.450	0.944
	3	44	0.682	0.055	0.543	0.778
Negative	Full	605	0.258	0.075	0.111	0.483
	1	165	0.245	0.058	0.130	0.409
	2	396	0.266	0.079	0.111	0.483
	3	44	0.244	0.085	0.115	0.426

Notes: descriptive statistics of all variables from 2007 to 2017. “Full” indicates the full sample, including 55 countries; “1” represents the high-income countries sample, including 15 countries; “2” represents the middle-income countries sample, including 36 countries; “3” represents the low-income countries sample, including 4 countries. The sample period is from 2007 to 2017.

**Table 3 ijerph-17-07530-t003:** Results of panel unit root test.

	LLC	Fisher-ADF	IPS	HT
Happiness	−44.413 ***	377.211 ***	−4.958 ***	−12.345 ***
Depression	−13.480 ***	273.052 ***	−4.469 ***	3.4738
Anxiety	−15.227 ***	383.485 ***	−9.783 ***	5.4507
National income	−28.981 ***	181.033 ***	−3.283 ***	2.8404
Unemployment	−17.470 ***	407.622 ***	−4.343 ***	−1.8569 **
Social support	−26.406 ***	357.420 ***	−6.125 ***	−13.802 ***
Freedom	−81.491 ***	292.300 ***	−3.743 ***	−11.030 ***
Generosity	−35.891 ***	268.514 ***	−3.540 ***	−10.478 ***
Positive	−13.150 ***	279.827 ***	−4.627 ***	−11.695 ***
Negative	−17.710 ***	329.668 ***	−3.958 ***	−11.816 ***

Notes: this table summarizes panel unit-root tests for happiness, depression, anxiety, national income, unemployment, social support, freedom (freedom to make life choices), generosity, positive affect (Positive) and negative affect (Negative) in 55 countries. LLC denotes Levin–Lin–Chu unit-root test; Fisher-ADF denotes Fisher type augmented Dickey–Fuller unit-root tests; IPS denotes Im–Pesaran–Shin unit-root test; HT denotes Harris Tzavalis unit-root test. *** and ** indicate significance at the 1% and 5% levels, respectively. The sample period is from 2007 to 2017.

**Table 4 ijerph-17-07530-t004:** The results of impact of national income on different types of mental health.

Model	Panel Tobit Model			Panel Regression Model		
Variable	Happiness	Depression	Anxiety	Happiness	Depression	Anxiety
National income	0.352 ***	0.233 ***	−0.051 ***	0.354 ***	0.217 ***	−0.078 ***
	(0.051)	(0.025)	(0.005)	(0.047)	(0.028)	(0.018)
Unemployment	−3.414 ***	0.880 ***	0.158	−3.299 ***	0.830 ***	0.078
	(0.865)	(0.176)	(0.103)	(0.856)	(0.180)	(0.116)
Social support	1.364 ***	0.138 **	0.134 ***	1.432 ***	0.131 **	0.129 ***
	(0.359)	(0.065)	(0.040)	(0.356)	(0.065)	(0.042)
Freedom	0.525 **	0.168 ***	0.225 ***	0.514 **	0.175 ***	0.231 ***
	(0.204)	(0.037)	(0.023)	(0.206)	(0.037)	(0.024)
Generosity	0.112	−0.166 ***	−0.081 ***	0.095	−0.165 ***	−0.083 ***
	(0.189)	(0.034)	(0.022)	(0.190)	(0.035)	(0.022)
Positive	0.994 ***	0.079	−0.087 **	1.047 ***	0.086	−0.098 **
	(0.343)	(0.063)	(0.040)	(0.341)	(0.064)	(0.041)
Negative	−0.623 *	0.334 ***	0.280 ***	−0.567	0.338 ***	0.282 ***
	(0.357)	(0.064)	(0.040)	(0.356)	(0.064)	(0.041)
_cons	3.335 ***	2.685 ***	3.318 ***	3.223 ***	2.701 ***	3.724 ***
	(0.428)	(0.128)	(0.045)	(0.414)	(0.086)	(0.056)
sigma_u	0.622 ***	0.714 ***	1.382 ***	—	0.726	1.376
	(0.063)	(0.068)	(0.132)	—	—	—
sigma_e	0.316 ***	0.055 ***	0.036 ***	—	0.055	0.036
	(0.010)	(0.002)	(0.001)	—	—	—
rho	0.795	0.994	0.999	—	0.994	0.999
N	605	605	605	605	605	605
Wald chi2	152.43 ***	212.25 ***	276.43 ***	—	—	—
LR test	621.03 ***	2468.36 ***	3489.92 ***	—	—	—
R^2^	—	—	—	—	0.255	0.281

Notes: this is a regression with panel tobit model and panel regression model for a balanced panel explaining the relationship between national income and different types of mental health from 2007 to 2017. ***, ** and * indicate significance at the 1%, 5% and 10% levels, respectively. Standard errors are in parentheses.

**Table 5 ijerph-17-07530-t005:** The impact of national income on mental health among high-income countries.

	Happiness		Depression		Anxiety	
GNI	0.059	0.323 *	0.198 ***	0.533 ***	−0.030 ***	0.099 ***
	(0.054)	(0.191)	(0.031)	(0.069)	(0.005)	(0.019)
GNI^2^		−0.038		−0.042 ***		−0.021 ***
		(0.026)		(0.008)		(0.003)
Unemp	−3.110 ***	−3.197 ***	0.932 ***	0.989 ***	0.528 ***	0.450 ***
	(0.795)	(0.791)	(0.239)	(0.220)	(0.124)	(0.106)
Soc	3.292 ***	3.176 ***	−0.137	−0.192	0.184	0.128
	(0.883)	(0.881)	(0.247)	(0.227)	(0.142)	(0.139)
Free	0.537	0.477	−0.036	−0.169 *	0.157 ***	0.105 **
	(0.334)	(0.336)	(0.093)	(0.089)	(0.054)	(0.047)
Gene	1.157 ***	1.106 ***	0.147 *	0.108	−0.009	−0.045
	(0.269)	(0.270)	(0.076)	(0.070)	(0.044)	(0.042)
Posi	1.414 ***	1.418 ***	0.128	0.062	0.016	−0.044
	(0.469)	(0.464)	(0.131)	(0.121)	(0.077)	(0.073)
Nega	−1.777 ***	−1.871 ***	0.309 **	0.256 *	−0.091	−0.082
	(0.550)	(0.545)	(0.147)	(0.135)	(0.086)	(0.084)
_cons	2.589 ***	2.412 ***	3.405 ***	3.086 ***	4.350 ***	4.327 ***
	(0.811)	(0.819)	(0.307)	(0.297)	(0.132)	(0.126)
sigma_u	0.348 ***	0.368 ***	0.641 ***	0.653 ***	1.435 ***	1.457 ***
	(0.069)	(0.075)	(0.117)	(0.120)	(0.262)	(0.266)
sigma_e	0.197 ***	0.195 ***	0.052 ***	0.048 ***	0.031 ***	0.030 ***
	(0.011)	(0.011)	(0.003)	(0.003)	(0.002)	(0.002)
rho	0.757	0.781	0.993	0.995	0.999	0.999
*N*	165	165	165	165	165	165
Wald chi2	147.47 ***	150.77 ***	61.14 ***	100.28 ***	63.82 ***	213.47 ***
LR test	128.10 ***	120.51 ***	679.86 ***	705.35 ***	1012.38 ***	1023.45 ***

Notes: this is a regression with panel Tobit model explaining the relationship between national income and different types of mental health among different countries with high income from 2007 to 2017. GNI and GNI^2^ refer to national income and its square value, respectively. ***, ** and * indicate significance at the 1%, 5% and 10% levels, respectively. Standard errors are in parentheses.

**Table 6 ijerph-17-07530-t006:** The impact of national income on mental health among middle-income countries.

	Happiness		Depression		Anxiety	
GNI	1.267 ***	2.723 ***	0.410 ***	1.609 ***	0.046	0.475 ***
	(0.222)	(0.615)	(0.060)	(0.140)	(0.040)	(0.102)
GNI^2^		−1.086 **		−0.746 ***		−0.266 ***
		(0.430)		(0.080)		(0.058)
Unemp	−5.540 ***	−5.966 ***	0.382	0.365	0.003	0.001
	(1.269)	(1.263)	(0.265)	(0.239)	(0.178)	(0.174)
Soc	0.688	0.650	0.128 *	0.153 **	0.147 ***	0.156 ***
	(0.432)	(0.427)	(0.073)	(0.066)	(0.049)	(0.048)
Free	0.546 **	0.565 **	0.194 ***	0.171 ***	0.205 ***	0.196 ***
	(0.257)	(0.255)	(0.044)	(0.040)	(0.029)	(0.029)
Gene	0.005	0.132	−0.253 ***	−0.181 ***	−0.122 ***	−0.097 ***
	(0.238)	(0.241)	(0.040)	(0.036)	(0.026)	(0.026)
Posi	0.908 **	0.539	0.131 *	−0.054	−0.107 **	−0.173 ***
	(0.437)	(0.456)	(0.078)	(0.073)	(0.052)	(0.053)
Nega	−0.807 *	−0.859 *	0.354 ***	0.276 ***	0.252 ***	0.224 ***
	(0.480)	(0.477)	(0.082)	(0.074)	(0.055)	(0.054)
_cons	3.767 ***	3.793 ***	2.501 ***	2.359 ***	3.255 ***	3.204 ***
	(0.523)	(0.511)	(0.153)	(0.146)	(0.178)	(0.176)
sigma_u	0.490 ***	0.477 ***	0.723 ***	0.704 ***	0.997 ***	0.984 ***
	(0.063)	(0.061)	(0.085)	(0.083)	(0.118)	(0.116)
sigma_e	0.335 ***	0.333 ***	0.054 ***	0.048 ***	0.036 ***	0.035 ***
	(0.013)	(0.012)	(0.002)	(0.002)	(0.001)	(0.001)
rho	0.681	0.672	0.995	0.995	0.999	0.999
*N*	396	396	396	396	396	396
Wald chi2	92.84 ***	103.15 ***	188.00 ***	318.14 ***	147.37 ***	176.11 ***
LR test	287.03 ***	283.76 ***	1552.76 ***	1606.55 ***	2201.83 ***	2221.22 ***

Notes: this is a regression with panel Tobit model explaining the relationship between national income and different types of mental health among different countries with middle income from 2007 to 2017. GNI and GNI^2^ refer to national income and its square value, respectively. ***, ** and * indicate significance at the 1%, 5% and 10% levels, respectively. Standard errors are in parentheses.

**Table 7 ijerph-17-07530-t007:** The impact of national income on mental health among low-income countries.

	Happiness		Depression		Anxiety	
GNI	−29.780 ***	−45.610	3.851 **	−2.473	4.758 ***	17.560 ***
	(8.804)	(52.360)	(1.633)	(9.434)	(0.337)	(3.788)
GNI2		117.100		42.990		−79.390 ***
		(379.900)		(63.18)		(25.340)
Unemp	34.750 ***	36.180 ***	2.519 *	2.689 *	−0.408	−0.365
	(10.68)	(11.46)	(1.438)	(1.448)	(0.611)	(0.581)
Soc	1.369	1.457	−0.089	−0.066	0.019	−0.018
	(0.915)	(0.954)	(0.123)	(0.127)	(0.055)	(0.051)
Free	0.565	0.520	0.079	0.074	0.089 **	0.068 *
	(0.627)	(0.643)	(0.087)	(0.086)	(0.035)	(0.035)
Gene	0.779	0.787	−0.144 *	−0.147 *	−0.001	0.015
	(0.622)	(0.619)	(0.083)	(0.082)	(0.037)	(0.033)
Posi	−1.499	−1.423	−0.100	−0.0921	0.0463	0.0634
	(1.023)	(1.044)	(0.138)	(0.137)	(0.059)	(0.055)
Nega	2.833 ***	3.002 ***	−0.004	0.060	0.276 ***	0.123 *
	(0.899)	(1.052)	(0.126)	(0.157)	(0.0536)	(0.0630)
_cons	4.171 ***	4.511 ***	2.657 ***	2.831 ***	3.026 ***	2.203 ***
	(1.175)	(1.618)	(0.282)	(0.385)	(0.064)	(0.215)
sigma_u	0.323 **	0.339 **	0.446 ***	0.461 ***	0.582 ***	0.351 ***
	(0.138)	(0.153)	(0.158)	(0.165)	(0.206)	(0.125)
sigma_e	0.294 ***	0.292 ***	0.039 ***	0.039 ***	0.017 ***	0.015 ***
	(0.033)	(0.033)	(0.004)	(0.004)	(0.002)	(0.002)
rho	0.547	0.573	0.993	0.993	0.999	0.998
*N*	44	44	44	44	44	44
Wald chi2	25.12 ***	25.38 ***	32.37 ***	33.40 ***	843.44 ***	491.11 ***
LR test	8.68 ***	6.94 ***	138.03 ***	114.03 ***	184.17 ***	173.26 ***

Notes: this is a regression with panel Tobit model explaining the relationship between national income and different types of mental health among different countries with low income from 2007 to 2017. GNI and GNI2 refer to national income and its square value, respectively. ***, ** and * indicate significance at the 1%, 5% and 10% levels, respectively. Standard errors are in parentheses.

**Table 8 ijerph-17-07530-t008:** The moderating effect between national income and happiness.

Happiness	(1)	(2)	(3)	(4)	(5)	(6)
National income	0.362 ***	0.333 ***	0.368 ***	0.326 ***	0.352 ***	0.341 ***
	(0.052)	(0.062)	(0.056)	(0.053)	(0.051)	(0.052)
NI * unemployment	0.710					
	(0.582)					
NI * Socialsupport		0.227				
		(0.406)				
NI * Freedom			−0.129			
			(0.187)			
NI * Generosity				0.225		
				(0.139)		
NI * Positive					−0.019	
					(0.244)	
NI * Negative						−0.329
						(0.283)
Unemployment	−3.789 ***	−3.435 ***	−3.478 ***	−3.466 ***	−3.408 ***	−3.229 ***
	(0.920)	(0.866)	(0.870)	(0.864)	(0.869)	(0.879)
Social support	1.327 ***	1.540 ***	1.341 ***	1.382 ***	1.365 ***	1.411 ***
	(0.360)	(0.476)	(0.360)	(0.358)	(0.359)	(0.360)
Freedom	0.530 ***	0.526 ***	0.450 *	0.519 **	0.524 **	0.499 **
	(0.204)	(0.204)	(0.231)	(0.204)	(0.204)	(0.205)
Generosity	0.096	0.110	0.113	0.188	0.111	0.111
	(0.189)	(0.189)	(0.189)	(0.195)	(0.190)	(0.189)
Positive	0.930 ***	0.973 ***	0.996 ***	1.010 ***	0.991 ***	1.010 ***
	(0.346)	(0.345)	(0.343)	(0.342)	(0.344)	(0.343)
Negative	−0.678 *	−0.630 *	−0.656 *	−0.607 *	−0.621 *	−0.777 **
	(0.359)	(0.357)	(0.360)	(0.356)	(0.358)	(0.380)
_cons	3.631 ***	4.896 ***	4.197 ***	3.726 ***	4.504 ***	3.557 ***
	(0.434)	(0.326)	(0.443)	(0.433)	(0.345)	(0.407)
sigma_u	0.629 ***	0.627 ***	0.620 ***	0.620 ***	0.621 ***	0.625 ***
	(0.064)	(0.064)	(0.063)	(0.063)	(0.064)	(0.063)
sigma_e	0.315 ***	0.315 ***	0.316 ***	0.315 ***	0.316 ***	0.315 ***
	(0.009)	(0.009)	(0.009)	(0.009)	(0.009)	(0.009)
*N*	605	605	605	605	605	605
Wald chi2	152.93 ***	151.83 ***	153.29 ***	155.94 ***	152.55 ***	153.32 ***
LR test	622.52 ***	614.48 ***	621.35 ***	614.35 ***	618.62 ***	617.47 ***

Notes: this is a regression with panel Tobit model explaining the moderating effect between national income and happiness from 2007 to 2017. NI represents national income. ***, ** and * indicate significance at the 1%, 5% and 10% levels, respectively. Standard errors are in parentheses.

**Table 9 ijerph-17-07530-t009:** The moderating effect between national income and depression.

Depression	(1)	(2)	(3)	(4)	(5)	(6)
National income	0.255 ***	0.243 ***	0.264 ***	0.326 ***	0.233 ***	0.233 ***
	(0.027)	(0.026)	(0.026)	(0.053)	(0.026)	(0.025)
NI * unemployment	0.294 ***					
	(0.113)					
NI * Socialsupport		−0.242 ***				
		(0.073)				
NI * Freedom			−0.149 ***			
			(0.035)			
NI * Generosity				0.225		
				(0.139)		
NI * Positive					0.005	
					(0.045)	
NI * Negative						−0.051
						(0.051)
Unemployment	0.740 ***	0.873 ***	0.831 ***	−3.466 ***	0.880 ***	0.917 ***
	(0.183)	(0.174)	(0.173)	(0.864)	(0.176)	(0.179)
Social support	0.123 *	−0.047	0.115 *	1.382 ***	0.137 **	0.145 **
	(0.064)	(0.085)	(0.064)	(0.358)	(0.065)	(0.065)
Freedom	0.166 ***	0.168 ***	0.074 *	0.519 **	0.168 ***	0.164 ***
	(0.037)	(0.037)	(0.043)	(0.204)	(0.037)	(0.037)
Generosity	−0.169 ***	−0.167 ***	−0.167 ***	0.188	−0.166 ***	−0.167 ***
	(0.034)	(0.034)	(0.034)	(0.195)	(0.034)	(0.034)
Positive	0.059	0.097	0.081	1.010 ***	0.080	0.083
	(0.064)	(0.063)	(0.062)	(0.342)	(0.064)	(0.063)
Negative	0.313 ***	0.341 ***	0.297 ***	−0.607 *	0.334 ***	0.310 ***
	(0.064)	(0.063)	(0.063)	(0.356)	(0.064)	(0.068)
_cons	3.057 ***	3.091 ***	3.136 ***	3.726 ***	3.029 ***	3.047 ***
	(0.123)	(0.112)	(0.123)	(0.433)	(0.113)	(0.121)
sigma_u	0.709 ***	0.715 ***	0.708 ***	0.620 ***	0.714 ***	0.714 ***
	(0.068)	(0.069)	(0.068)	(0.063)	(0.069)	(0.069)
sigma_e	0.055 ***	0.055 ***	0.054 ***	0.315 ***	0.055 ***	0.055 ***
	(0.002)	(0.002)	(0.002)	(0.009)	(0.002)	(0.002)
*N*	605	605	605	605	605	605
Wald chi2	221.77 ***	226.58 ***	237.12 ***	218.73 ***	212.28 ***	213.66 ***
LR test	2470.66 ***	2476.55 ***	2484.70 ***	2461.54 ***	2456.97 ***	2466.76 ***

Notes: this is a regression with panel Tobit model explaining the moderating effect between national income and depression from 2007 to 2017. NI represents national income. ***, ** and * indicate significance at the 1%, 5% and 10% levels, respectively. Standard errors are in parentheses.

**Table 10 ijerph-17-07530-t010:** The moderating effect between national income and anxiety.

Anxiety	(1)	(2)	(3)	(4)	(5)	(6)
National income	−0.051 ***	−0.003	−0.048 ***	−0.056 ***	0.077 ***	−0.061 ***
	(0.009)	(0.005)	(0.008)	(0.007)	(0.004)	(0.010)
NI * unemployment	0.084					
	(0.071)					
NI* Socialsupport		0.083 *				
		(0.045)				
NI* Freedom			−0.036			
			(0.023)			
NI* Generosity				0.005		
				(0.017)		
NI* Positive					0.100 ***	
					(0.029)	
NI* Negative						−0.164 ***
						(0.032)
Unemployment	0.099	0.348 ***	0.134	0.144	0.505 ***	0.239 **
	(0.113)	(0.085)	(0.104)	(0.104)	(0.089)	(0.111)
Social support	0.129 ***	0.198 ***	0.128 ***	0.133 ***	0.129 ***	0.156 ***
	(0.041)	(0.054)	(0.041)	(0.041)	(0.039)	(0.041)
Freedom	0.226 ***	0.211 ***	0.203 ***	0.226 ***	0.203 ***	0.214 ***
	(0.023)	(0.023)	(0.027)	(0.023)	(0.024)	(0.023)
Generosity	−0.082 ***	−0.085 ***	−0.081 ***	−0.079 ***	−0.060 ***	−0.083 ***
	(0.022)	(0.021)	(0.022)	(0.023)	(0.022)	(0.022)
Positive	−0.095 **	−0.081 **	−0.088 **	−0.088 **	−0.009	−0.077 **
	(0.040)	(0.039)	(0.040)	(0.040)	(0.041)	(0.039)
Negative	0.275 ***	0.278 ***	0.271 ***	0.281 ***	0.255 ***	0.203 ***
	(0.041)	(0.041)	(0.041)	(0.041)	(0.043)	(0.043)
_cons	3.279 ***	2.598 ***	3.441 ***	3.265 ***	3.532 ***	3.523 ***
	(0.048)	(0.037)	(0.048)	(0.047)	(0.036)	(0.050)
sigma_u	1.382 ***	1.698 ***	1.381 ***	1.384 ***	1.210 ***	1.346 ***
	(0.132)	(0.162)	(0.132)	(0.132)	(0.115)	(0.129)
sigma_e	0.035 ***	0.036 ***	0.035 ***	0.035 ***	0.037 ***	0.035 ***
	(0.001)	(0.001)	(0.001)	(0.001)	(0.001)	(0.001)
*N*	605	605	605	605	605	605
Wald chi2	235.39 ***	291.76 ***	265.11 ***	259.16 ***	1429.09 ***	286.56 ***
LR test	3485.76 ***	3449.53 ***	3492.51 ***	3490.26 ***	3436.54 ***	3513.42 ***

Notes: this is a regression with panel Tobit model explaining the moderating effect between national income and anxiety from 2007 to 2017. NI represents national income. ***, ** and * indicate significance at the 1%, 5% and 10% levels, respectively. Standard errors are in parentheses.

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
