# Peer review of "Are There Heterogeneous Impacts of National Income on Mental Health?"

_ijerph, 2020, doi:10.3390/ijerph17207530_

Round 1
Reviewer 1 Report
I consider the topic of the article to be original and useful for practical use, especially by political representations.
I consider the scope of empirical surveys and the research plan itself to be particularly beneficial.
I consider the generality of the conclusions in the sense of "different causes cause different results" as well as brief recommendations for use in practice to be weaker. However, these comments do not diminish the benefits of the published results. Therefore, I recommend the article for publication.
In addition, the fundamental criteria are evaluated by a scale in the system and I do not consider it necessary to comment on them again in the text part. Of course, I can do it if you deem it necessary.
Author Response
Dear Editors and Reviewers:
Thank you for your letter and for the reviewers’ comments concerning our manuscript entitled “Are there Heterogeneous Impacts of National Income on Mental Health?”(ijerph_917836). Those comments are all valuable and very helpful for revising and improving our paper, as well as the important guiding significance to our researches. We have studied comments carefully and have made corrections which we hope meet with approval. The revised portion is marked in red in the paper. The main corrections in the paper and the response to the reviewer’s comments are as following:
Responds to the reviewer’s comments:
Reviewer 1:
I consider the topic of the article to be original and useful for practical use, especially by political representations.
I consider the scope of empirical surveys and the research plan itself to be particularly beneficial.
I consider the generality of the conclusions in the sense of "different causes cause different results" as well as brief recommendations for use in practice to be weaker. However, these comments do not diminish the benefits of the published results. Therefore, I recommend the article for publication.
In addition, the fundamental criteria are evaluated by a scale in the system and I do not consider it necessary to comment on them again in the text part. Of course, I can do it if you deem it necessary.
Response:
Thanks for your valuable suggestions. According to your comments, we make the following changes. First, we strengthen our recommendations for use in practice in line 529-536 on page 20, which have marked red. In addition, we deleted the fundamental criteria in table 1 on Page 6.
This manuscript has been edited and proofread according to your valuable comments.
We hope that the revised version of the manuscript is now acceptable for publication.
I look forward to hearing from you soon.
With best wishes,
Yours sincerely,
Tinghui Li

Reviewer 2 Report
This paper discusses the heterogeneous income impact on different mental health types. The manuscript presents some problems that must be addressed before publication.
1) A lot of spelling, punctuation, and grammar errors. I suggest the authors a professional proofreading service.
2) The manuscript organization needs to be improved. A section on related work should be added for purposes of comparison and better observation of the main contributions of the study.
3) Figures must be improved.
4) Include suggestions for future works and study limitations.
Author Response
Dear Editors and Reviewers:
Thank you for your letter and for the reviewers’ comments concerning our manuscript entitled “Are there Heterogeneous Impacts of National Income on Mental Health?”(ijerph_917836). Those comments are all valuable and very helpful for revising and improving our paper, as well as the important guiding significance to our researches. We have studied comments carefully and have made corrections which we hope meet with approval. The revised portion is marked in red in the paper. The main corrections in the paper and the response to the reviewer’s comments are as following:
Responds to the reviewer’s comments:
Reviewer 2:
This paper discusses the heterogeneous income impact on different mental health types. The manuscript presents some problems that must be addressed before publication.
- A lot of spelling, punctuation, and grammar errors. I suggest the authors a professional proofreading service.
Response: We are very sorry for our incorrect writing and we use a professional proofreading service to complete the modification. For example,
- P. 1, line 11: “Based” instead of “Therefore, based”
- P. 1, line 13: “it analyzes” instead of “this paper explores”
- P. 1, line 14: “the heterogeneous moderating effects of national income on mental health mechanisms are elaborated as well.” instead of “this paper investigates the heterogeneous moderating effects on national income-mental health mechanisms.”
- P. 1, line 38: “losing” instead of “lost”
- P. 1, line 39-42: “there is a growing recognition that the mental health plays an important role in achieving global development goals, as well as illstrating by the inclusion of mental health in the Sustainable Development Goals.” instead of “there has been increasing acknowledgement of important role mental health plays in achieving global development goals, as illustrated by the inclusion of mental health in the Sustainable Development Goals.”
- P. 4, line 142: “help them to reduce anxiety” instead of “reducing their anxiety”
In addition to the above modifications, there are some modifications in the rest of paper, which have been marked red in the text.
- The manuscript organization needs to be improved. A section on related work should be added for purposes of comparison and better observation of the main contributions of the study.
Response: Thanks for reviewer’s suggestions. According to your comments, we add the related work in the part of introduction in line 75-85 on page 2.
- Figures must be improved.
Response: Thanks for your concerns. According to your comments, we improve the figure 2 on Page 8.
- Include suggestions for future works and study limitations.
Response: Thanks for your comments. We add future works and study limitations in line 541-552 on page 20.
This manuscript has been edited and proofread according to your valuable comments.
We hope that the revised version of the manuscript is now acceptable for publication.
I look forward to hearing from you soon.
With best wishes,
Yours sincerely,
Tinghui Li

Reviewer 3 Report
I carefully read through this manuscript. This manuscript still has some flaws particularly in the method and result sections. And some problems need to be clarified before it reaches the standard of publication.
(1) This manuscript seems to be a data driven study, which seriously lacks theoretical foundation. From the results of Table 4, "national income" positively affects depression and negatively affects anxiety at the same time. Is there any theory or evidence of prior study can support such findings? I cannot see any relevant explanation or empirical support from the prior literature.
(2) Even from the section 2.3 (variable and data source) and the so-called detailed description of variables, I cannot see any necessary information trying to clarify the exact definition of the key dependent variables (anxiety, depression). What's the items used in the survey. Different scales can yield very different results, and the constructs (anxiety, depression) have many scales. Can these items partially explain the inexplicable results of Table 4?
(3) Why the authors just choose these two unit root tests? If it is out of robustness, more tests are still in need (e.g., the Im Shin Pasaran test, Hadri test).
(4) The results of Table 5 are more serious. The authors need to carefully reconsider why the supplement of quadratic term of "anxiety" (high income group) can change the regression result of the linear term from -0.030 (***) to 0.099 (***) . The abrupt change from negatively significant to positively significant is very unusual. The results show some sensitivity. Moreover, in the middle income group, the supplement of quadratic term of "anxiety" makes the coefficient of linear term from insignificant to significant (***). Besides, in the low income group, the supplement of quadratic term of "depression" makes the coefficient of linear term from significant (***) to insignificant. There are too many problems need to be clarified. In the Table 5, the coefficients of other variables need to be shown in details, instead of being shown as "included". The correlation matrix among variables also need to be shown as a Appendix section, in order to show that the results are not sensitive, and affected by the inter-correlation among variables or collinearity.
(5) The results of Table 6 must be reshaped into a readable form. The current form is very unclear. The regression must be shown as the interaction term instead of the current layout. The coefficients of other variables should be displayed in details. If the space is limited, the table can be separated into several parts.
(6)It needs to be emphasized that, the current manuscript is still a data driven manuscript, and the theoretical support of the inexplicable finding as mentioned above is quite week. The authors should spend substantial length on the review of prior studies, looking for enough support to sustain the current empirical findings.
Author Response
Dear Editors and Reviewers:
Thank you for your letter and for the reviewers’ comments concerning our manuscript entitled “Are there Heterogeneous Impacts of National Income on Mental Health?”(ijerph_917836). Those comments are all valuable and very helpful for revising and improving our paper, as well as the important guiding significance to our researches. We have studied comments carefully and have made corrections which we hope meet with approval. The revised portion is marked in red in the paper. The main corrections in the paper and the response to the reviewer’s comments are as following:
Responds to the reviewer’s comments:
Reviewer 3:
I carefully read through this manuscript. This manuscript still has some flaws particularly in the method and result sections. And some problems need to be clarified before it reaches the standard of publication.
(1) This manuscript seems to be a data driven study, which seriously lacks theoretical foundation. From the results of Table 4, "national income" positively affects depression and negatively affects anxiety at the same time. Is there any theory or evidence of prior study can support such findings? I cannot see any relevant explanation or empirical support from the prior literature.
Response: Thanks for your comment. Our theoretical foundation has been shown in the part of 2.1 hypotheses on page 4-5, which including three aspects. First, the heterogeneous impact of national income on mental health is mainly reflected in different types of mental health. Second, the heterogeneous effect of national income on different types of mental health is mainly reflected in countries with different income levels. Third, the heterogeneity of the influence mechanism of national income on mental health is mainly reflected in different types of mental health.
Besides that, we add some explanations of results of Table 4, and we add evidence of prior study to support such findings in line 319-329 on Page 11.
(2) Even from the section 2.3 (variable and data source) and the so-called detailed description of variables, I cannot see any necessary information trying to clarify the exact definition of the key dependent variables (anxiety, depression). What's the items used in the survey. Different scales can yield very different results, and the constructs (anxiety, depression) have many scales. Can these items partially explain the inexplicable results of Table 4?
Response: We are very sorry for our unclear definition of the key dependent variables (anxiety, depression). According to your comments, we add the exact definition of anxiety and depression to table 1 on page 6. The items used in the survey are the prevalence of depression disorders and anxiety disorders respectively. The prevalence of depression disorders or anxiety disorders is defined by the number of people in the sample with the characteristic of depression or anxiety disorders, divided by the total number of people in the sample.
(3) Why the authors just choose these two unit root tests? If it is out of robustness, more tests are still in need (e.g., the Im Shin Pasaran test, Hadri test).
Response: According to the Reviewer’s suggestion, we add two unit root tests (Im-Pesaran-Shin test and Harris Tzavalis test) to check the robustness on Page 9. The reason we do not use Hadri test is that our data are long panel data, which are not more suitable. Therefore, we added the other two unit root test (Im-Pesaran-Shin test and Harris Tzavalis test) to check the robustness.
(4) The results of Table 5 are more serious. The authors need to carefully reconsider why the supplement of quadratic term of "anxiety" (high income group) can change the regression result of the linear term from -0.030 (***) to 0.099 (***). The abrupt change from negatively significant to positively significant is very unusual. The results show some sensitivity. Moreover, in the middle income group, the supplement of quadratic term of "anxiety" makes the coefficient of linear term from insignificant to significant (***). Besides, in the low income group, the supplement of quadratic term of "depression" makes the coefficient of linear term from significant (***) to insignificant. There are too many problems need to be clarified. In the Table 5, the coefficients of other variables need to be shown in details, instead of being shown as "included". The correlation matrix among variables also need to be shown as a Appendix section, in order to show that the results are not sensitive, and affected by the inter-correlation among variables or collinearity.
Response: Thanks for your comments. According to your comments, we make the following modifications. Firstly, we show the coefficients of other variables in details in Table 5-7 on Page 11-15. Besides that, we show the correlation matrix among variables in Appendix section on Page 21-22.
Finally, for the regression that include quadratic terms:
y=β0+β1 x12+β2 x1+β3 x2+β4 x3+ε,
The significance of x1 is not so important, because the focus is the significance of x12. The purpose of quadratic terms is to focus on the nonlinear relationship between national income and metal health. The linear relationship is different from nonlinear relationship, and the significant of linear relationship is independent of the significance of the nonlinear relationship.
Therefore, firstly, the quadratic term of "anxiety" is actually considering the nonlinear effect of national income on anxiety. And the impact of national income on anxiety may be -0.021*national income2+0.099*national income, instead of 0.099.
Secondly, as for the middle income group, the significance of supplement of quadratic term of "anxiety" is independent from the significance of the coefficient of linear term.
Similarly, in the low income group, the significance of the supplement of quadratic term of "depression" is also independent from the significance of the coefficient of linear term.
(5) The results of Table 6 must be reshaped into a readable form. The current form is very unclear. The regression must be shown as the interaction term instead of the current layout. The coefficients of other variables should be displayed in details. If the space is limited, the table can be separated into several parts.
Response: Thanks for your valuable comments. According to your comments, we reshape Table 6 into three readable forms: Table 8-10 on Page 16-19.
(6)It needs to be emphasized that, the current manuscript is still a data driven manuscript, and the theoretical support of the inexplicable finding as mentioned above is quite week. The authors should spend substantial length on the review of prior studies, looking for enough support to sustain the current empirical findings.
Response: Thanks for your valuable comments. We reviewed prior studies and find enough support to sustain the current empirical findings. For example, national income plays a positive effect on depression in line 319-323 on Page 11.
This manuscript has been edited and proofread according to your valuable comments.
We hope that the revised version of the manuscript is now acceptable for publication.
I look forward to hearing from you soon.
With best wishes,
Yours sincerely,
Tinghui Li
